# FAK loss reduces BRAF[V600E]-induced ERK phosphorylation to promote intestinal stemness and cecal tumor formation

Chenxi Gao[1], Huaibin Ge[2], Shih-Fan Kuan[3], Chunhui Cai[4], Xinghua Lu[4], Farzad Esni[5], Robert E Schoen[1], Jing H Wang[2], Edward Chu[2]*[†], Jing Hu[1]*

[1]Department of Pharmacology and Chemical Biology, University of Pittsburgh School of Medicine, Pittsburgh, United States; [2]UPMC Hillman Cancer Center, Division of Hematology and Oncology, Department of Medicine, University of Pittsburgh, Pittsburgh, United States; [3]Department of Pathology, University of Pittsburgh School of Medicine, Pittsburgh, United States; [4]Department of Biomedical Informatics, University of Pittsburgh, Pittsburgh, United States; [5]Department of Surgery, University of Pittsburgh School of Medicine, Pittsburgh, United States

**\*For correspondence:**
edward.chu@einsteinmed.org
(EC);
huj3@upmc.edu (JH)

**Present address:** [†]Albert Einstein Cancer Center, Albert Einstein College of Medicine, Bronx, United States

**Abstract** *BRAF*[V600E] mutation is a driver mutation in the serrated pathway to colorectal cancers. BRAF[V600E] drives tumorigenesis through constitutive downstream extracellular signal-regulated kinase (ERK) activation, but high-intensity ERK activation can also trigger tumor suppression. Whether and how oncogenic ERK signaling can be intrinsically adjusted to a 'just-right' level optimal for tumorigenesis remains undetermined. In this study, we found that FAK (Focal adhesion kinase) expression was reduced in *BRAF*[V600E]-mutant adenomas/polyps in mice and patients. In *Vil1-Cre;BRAF*[LSL-V600E/+];*Ptk2*[fl/fl] mice, *Fak* deletion maximized BRAF[V600E]'s oncogenic activity and increased cecal tumor incidence to 100%. Mechanistically, our results showed that Fak loss, without jeopardizing BRAF[V600E]-induced ERK pathway transcriptional output, reduced EGFR (epidermal growth factor receptor)-dependent ERK phosphorylation. Reduction in ERK phosphorylation increased the level of Lgr4, promoting intestinal stemness and cecal tumor formation. Our findings show that a 'just-right' ERK signaling optimal for *BRAF*[V600E]-induced cecal tumor formation can be achieved via Fak loss-mediated downregulation of ERK phosphorylation.

## eLife assessment

In this **important** study, the authors use a genetically engineered mouse model to reveal a tumor suppressive role for focal adhesion kinase in right-sided colon cancer. The evidence in support of the authors' claims is generally **solid**, although the data supporting the mechanism through which FAK deletion promotes tumorigenesis are **incomplete**. This work will be of interest to cancer researchers and others studying the biological consequences of tuning signal transduction pathways.

## Introduction

Colorectal cancer (CRC) is a heterogeneous disease arising through several discrete evolutionary pathways. The best-known and most-studied pathway to CRC is the canonical pathway, in which cancer originates from conventional adenomatous polyps bearing *APC* (*adenomatous polyposis coli*)

mutation (*Powell et al., 1992*; *Cancer Genome Atlas, 2012*). Recently a new 'alternative' pathway through serrated adenoma—the serrated pathway—has been uncovered. Mice studies have established that the *BRAF*^V600E mutation is a driver mutation in the serrated pathway (*Rad et al., 2013*; *Carragher et al., 2010*; *Rustgi, 2013*). In patients, *BRAF*^V600E mutation is found in 50–67% of serrated CRC (*Lannagan et al., 2019*) and 10–15% of all CRCs (*Davies et al., 2002*).

The 'Goldilocks principle' applies to mutant *APC*-driven and mutant *BRAF*-driven intestinal tumorigenesis: a threshold of oncogenic signaling needs to be reached for dysplastic lesions to form, but optimum tumor development requires 'just-right' levels of oncogenic signaling, with too much being as detrimental as too little. In the canonical pathway to CRC, the primary driving force is mutant APC-mediated activation of Wnt/β-catenin signaling (*Morin et al., 1997*), and the 'just-right' level of Wnt/β-catenin signaling optimal for tumor formation is achieved mainly by the selection for specific APC mutant proteins based on their residual β-catenin-downregulating activity (*Albuquerque et al., 2002*; *Leedham et al., 2013*; *Christie et al., 2013*; *Buchert et al., 2010*). The selection for *APC* mutations in the intestine is influenced by the underlying basal/physiological level of Wnt activity and stemcell number, and *APC* mutation spectra vary throughout the intestinal tract resulting in different *APC* mutation spectra in the proximal and distal CRCs (*Leedham et al., 2013*; *Christie et al., 2013*). In addition to the different mutation spectra, the 'optimal' thresholds for proximal and distal cancers are also variable (*Christie et al., 2013*).

BRAF^V600E drives tumorigenesis through constitutive downstream ERK1/2 activation (*Wellbrock et al., 2004*), but hyperactivation of ERK induced by oncogenic BRAF^V600 is not tolerated in the intestine: high ERK activation, induced by transgenic expression of oncogenic BRAF (BRAF^V600K) or by activation of two BRAF alleles in *BRAF*^V600E/V600E mutant mice, engages tumor suppressive mechanisms, causing loss of stem cells and induction of differentiation and senescence (*Riemer et al., 2015*; *Tong et al., 2017*). Lowering ERK activation by treatment with ERK or MEK (mitogen-activated protein kinase kinase) inhibitor counteracted BRAF^V600E-induced organoid disintegration (*Riemer et al., 2015*; *Brandt et al., 2019*). It is therefore presumed that maintaining ERK activation within a narrow threshold range to avoid engaging tumor suppression is pivotal for mutant BRAF to exhibit the strongest transforming activity. However, despite being highly anticipated (*Brandt et al., 2019*), the existence of in vivo intrinsic fine-tuning of mutant *BRAF*-induced ERK activation has never been experimentally examined. Given that over 60 mutations have now been identified in *BRAF* (*Wellbrock et al., 2004*; *Zebisch and Troppmair, 2006*), theoretically, mutation selection could be a way to achieve optimal ERK activation. However, because the V600E mutation accounts for about 90% of BRAF mutation seen in human cancer (*Rajagopalan et al., 2002*), mutation selection is not the primary means to achieve the 'just-right' levels of oncogenic ERK signaling. Normally, ERK activation is self-limiting by the rapid inactivation of upstream kinases and delayed induction of dual-specific MAKP phosphatases (MKPs/DUSPs) *Lake et al., 2016*. Although feedback inhibitors of ERK signaling, including DUSPs are overexpressed in BRAF^V600E-expressing cells, the ERK signaling pathway is refractory to upstream feedback inhibition (*Pratilas et al., 2009*). EGFR is a core receptor upstream of the MAPK kinase axis. In vitro cell culture studies show that all activating BRAF mutants are RAS-independent (*Yao et al., 2015*): neither RAS inhibition (*Yao et al., 2015*) nor EGFR inhibition (*Corcoran et al., 2012*; *Prahallad et al., 2012*) was able to inhibit mutant-BRAF-induced ERK phosphorylation in *BRAF*-mutant human CRC cell lines.

In this study, we addressed whether BRAF^V600E-induced ERK activation is still tuneable during tumorigenesis in vivo. If yes, what are the factors involved in the regulation? Can BRAF^V600E-induced ERK activation be fine-tuned to a 'just-right' level optimal for tumor initiation? Our study identified FAK as a key regulator of BRAF^V600E-induced ERK activation in mutant *BRAF*-induced serrated tumor formation/initiation and revealed that FAK loss allows BRAF^V600E-induced ERK signaling to reach the permissive threshold 'just-right' for cecal tumors to form.

## Results
### FAK expression is reduced in *BRAF*^V600E-mutant serrated lesions in humans and mice

FAK is a cytoplasmic non-receptor tyrosine kinase involved in many aspects and types of cancer (*Sulzmaier et al., 2014*). To determine the role of FAK in mutant *BRAF*-induced serrated CRC, we first

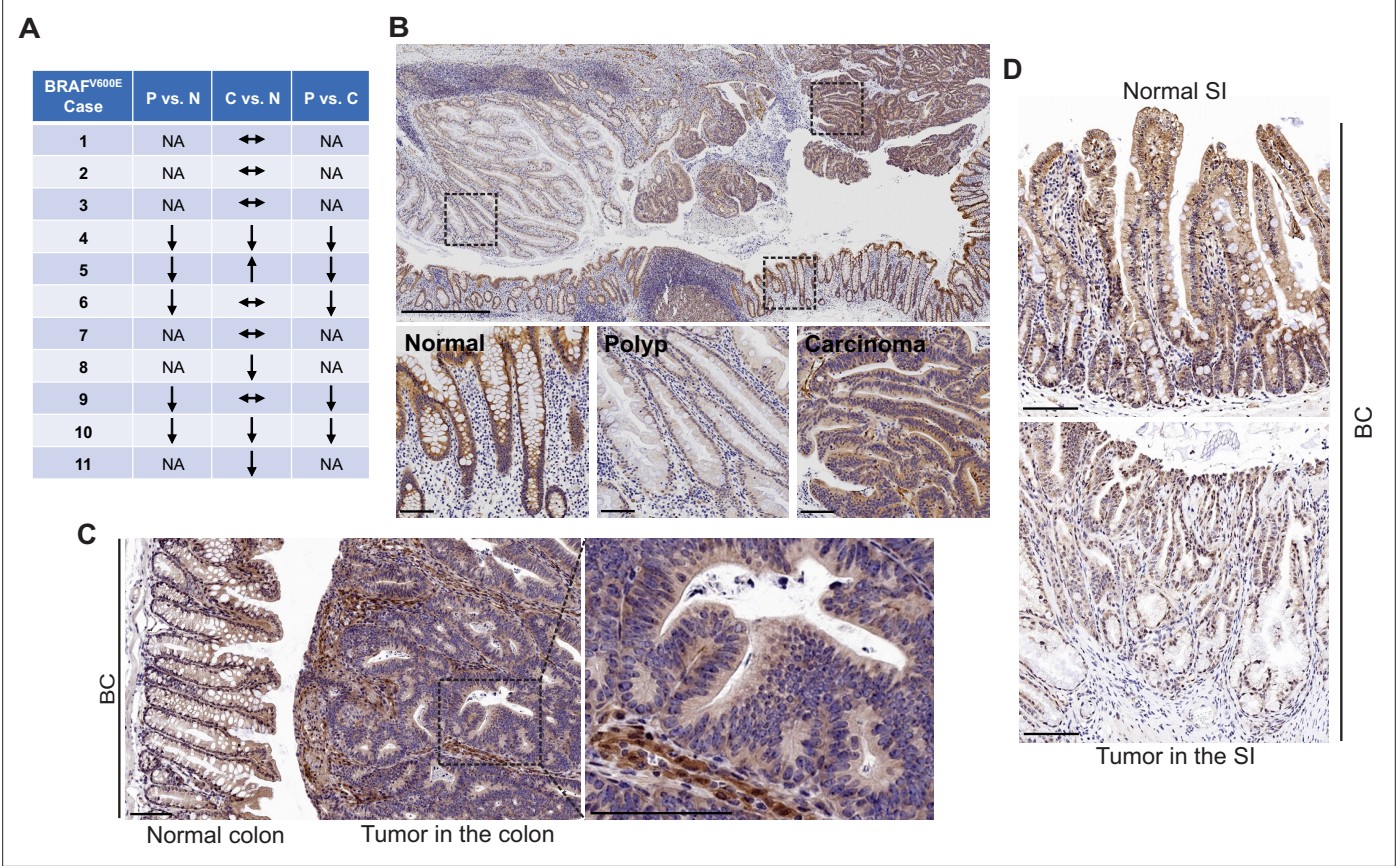

**Figure 1.** FAK downregulation in serrated tumors. (**A**) Summary of FAK IHC staining in 11 human *BRAF*^V600E-mutant CRC samples. N represents normal colon; P represents polyp; C represents carcinoma; NA, not applicable; ↔ represents no change; ↑ represents an increase. ↓ represents a decrease. (**B**) Representative IHC staining of *BRAF*^V600E-mutant patient SSA/P, serrated colorectal adenoma, and adjacent normal tissues. (**C**) IHC staining of Fak in small intestine tumors in a 12-month-old BC mouse. (**D**) Representative IHC staining of Fak in colon tumor in 12-month-old BC mice. Scale bars in (**B**) 1 mm (upper panel) and 100 µm (lower panel). Scale bars in (**C**) and (**D**) 100 µm.

evaluated FAK protein expressions in human *BRAF*^V600E-mutated serrated tumors (11 cases). We examined tissue sections containing *BRAF*^V600E-mutant CRCs, sessile serrated adenoma/polyps (SSA/P)s, and adjacent histologically normal colon from the same tissue block. Results of immunohistochemistry (IHC) staining showed that FAK protein levels were lower in SSA/Ps (5/5) than in normal intestines and CRCs (5/5) (*Figure 1A*). FAK expression was more complex in CRCs. FAK levels in CRCs were either similar to (6/11) or lower (4/11) or higher (1/11) than that of the normal intestines (*Figure 1A and B*). FAK was mainly localized in the cytoplasm (*Figure 1B*). In mice, compared to the neighboring normal mucosa or stroma in the tumor, Fak protein levels were substantially decreased in carcinomas in the colon (*Figure 1C*) and adenomas/polyps in the small intestine (SI; *Figure 1D*) in *Vil1-Cre;BRAF*^LSL-V600E/+ (BC) mice. The downregulation of FAK in human and mouse polyps suggests that FAK loss may play a role in *BRAF*^V600E-induced tumor formation/initiation.

## Fak deletion promotes *BRAF*^V600E-induced cecal tumor formation

Previous mice studies show that *Fak* deletion suppresses mammary tumorigenesis (*Pylayeva et al., 2009*; *Luo et al., 2009*), mutant *Apc*-induced intestinal tumorigenesis (*Ashton et al., 2010*), skin tumor formation (*McLean et al., 2004*), and hepatocarcinogenesis (*Shang et al., 2015*). To address the functional significance of FAK downregulation in *BRAF*^V600E-induced serrated tumor formation/initiation, we generated the *Vil1-Cre;BRAF*^LSL-V600E/+;*Ptk2*^fl/fl (FBC) mice. The Cre-mediated recombination efficiency was confirmed by tdTomato-reporter expression in intestinal crypts in *Vil1-Cre;Rosa26*^LSL-tdTomato/+ mice (*Figure 2—figure supplement 1A*). Deletion of Fak in the intestinal epithelium was further confirmed by IHC staining of the intestine in FBC mice (*Figure 2—figure supplement 1B*).

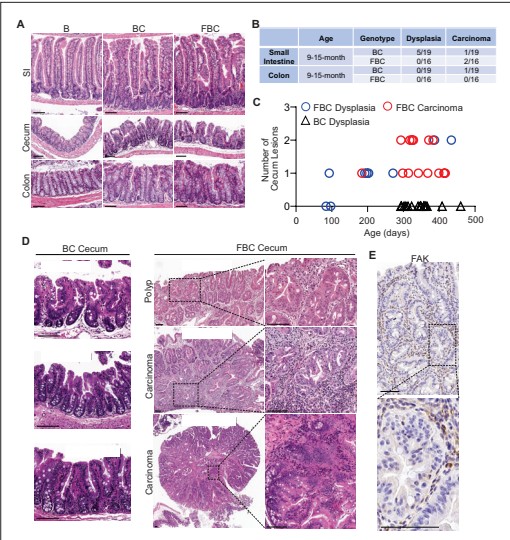

**Figure 2.** *Fak* loss enhances *BRAF*^V600E-driven cecal tumorigenesis in mice. (**A**) Representative hematoxylin and eosin (H&E) staining of the small intestine, cecum, and colon from indicated 6-week-old mice. (**B**) Summary of tumor incidence at small intestine and colon in indicted mice at the indicated age. (**C**) Summary of tumor incidence and tumor stage at cecum in indicated mice at the indicated age. (**D**) H&E staining of the cecum in BC mice and cecal serrated adenoma/polyp and carcinoma in FBC mice at the indicated age. (**E**) Representative IHC staining of Fak in cecal tumors in FBC mice. Scale bars: 100 μm.

The online version of this article includes the following figure supplement(s) for figure 2:

**Figure supplement 1.** Fak deletion promotes *BRAF*^V600E-induced cecal tumorigenesis.

Similar to that seen in BC mice, compared to the *BRAF*^LSL-V600E/+ (B) mice, the FBC mice exhibited hyperplasia throughout the intestine (*Figure 2A*) and thickened small and large intestines (*Figure 2—figure supplement 1C*). In BC mice, intestinal tumors were primarily developed in the small intestine at nine months or older (*Figure 2B*). *Fak* loss had minimal impact on tumor incidence in the small intestine and the colon; however, it greatly enhanced BRAF^V600E-induced cecal tumor formation: cecal tumor incidence increased from 0% (0/15) in 9-month or older BC mice to 100% (16/16) in FBC mice (*Figure 2C*). Cecal adenoma/polyp started to develop in 3-month FBC mice, and after 6 months, all mice (4/4) developed cecal tumors, and 25% of the tumors (1/4) were carcinomas (*Figure 2C and D*). At nine months or older, 100% of the mice developed cecal tumors with a high incidence (13/16) of carcinoma (*Figure 2C and D*, *Figure 2—figure supplement 1D*). IHC staining confirmed that while the stroma showed strong Fak staining, tumor cells were Fak negative (*Figure 2E*), hence validating that tumors were originated from Fak-deleted epithelial cells. Of note, no tumor metastasis was found in FBC mice. FBC mice were aged up to 434 days, and the life span of FBC mice was similar to that of BC mice.

Together, these results revealed that *Fak* deletion promotes, rather than inhibits, *BRAF*^V600E-induced cecal tumor formation. *BRAF*-mutant CRCs are primarily located in the right colon, including the cecum (*Clarke and Kopetz, 2015*). The same primary tumor location suggests that the FBC model truthfully recapitulates human *BRAF*-mutant serrated CRCs, at least by location.

## The molecular feature of the cecal tumors in FBC mice closely resembles human SSA/Ps

To characterize the molecular signatures of the cecal tumor in FBC mice, we performed whole-exome sequencing on paired tumors (n=2) and neighboring mucosa. No additional driver mutations were detected in the cecal tumors (*Supplementary file 1*), implying that cecal tumor formation in FBC mice does not require additional driver mutations. To evaluate the relevance of FBC cecal tumors to humans, we performed RNA-sequencing (RNA-seq) and Gene Set Enrichment Analysis (GSEA) to determine whether FBC cecal tumors exhibited similar gene expression signatures as human SSA/Ps (*Kanth et al., 2016*). The results showed that upregulated genes in human SSA/Ps were significantly enriched in cecal tumors in FBC mice (*Figure 3A*). Downregulated genes in human SSA/P were also reduced in FBC tumors (*Figure 3B*). Together, these results suggest that the FBC cecal tumors greatly resemble human serrated lesions at the molecular level.

About 50% of *BRAF*-mutated CRCs exhibit defective DNA mismatch repair (*Rajagopalan et al., 2002*). The results of microsatellite instability (MSI) analysis indicated that most FBC cecal tumors were microsatellite stable (MSS; *Figure 3C*). It has been shown that mismatch repair deficiency accelerates *BRAF*-driven serrated tumorigenesis (*Tong et al., 2021*). Maximizing the oncogenic activity of BRAF^V600E without mismatch repair gene mutation and additional driver mutations suggests that in FBC mice, Fak loss created a 'just-right' environment optimal for MSS serrated cecal tumor to form.

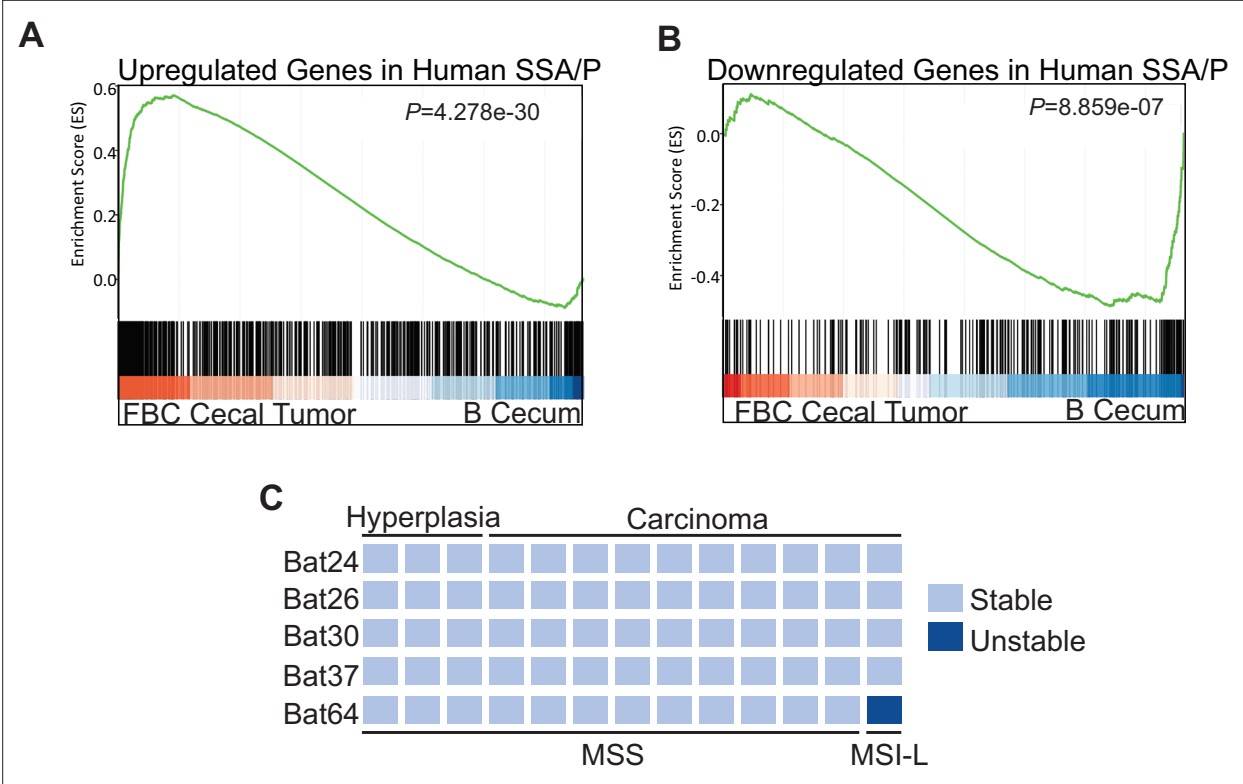

**Figure 3.** Molecular characterization of cecal tumors in FBC mice. (**A**) GSEA plot showing enrichment of human SSA/Ps signature genes (upregulated genes in SSA/Ps) in FBC cecal tumors vs normal cecal mucosa of B mice. (**B**) GSEA plot showing that downregulated genes in human SSA/Ps were also reduced in FBC cecal tumors. (**C**) Microsatellite instability status of FBC mice cecal mucosa and cecal carcinomas. Each column represents one sample.

## Fak loss increases intestinal stemness by upregulating Lgr4 levels in FBC mice

We explored the molecular mechanism underlying *Fak* loss-enhanced cecal tumor formation. Consistent with a prior report (*Ashton et al., 2010*), we did not detect any abnormalities in the intestine in *Vil1-Cre; Ptk2*$^{fl/fl}$ mice, implying that FAK loss by itself is not a driving force for intestinal tumorigenesis. A prior study showed that upon TGFβ (transforming growth factor β) receptor inactivation, *BRAF*$^{V600E}$-induced right-sided tumorigenesis is supported by microbial-driven inflammation (*Leach et al., 2021*). To test the role of inflammation in FBC tumor formation, we compared sub-cryptal proprial neutrophil infiltration using myeloperoxidase (MPO) as a neutrophil marker for IHC staining. The results showed that consistent with prior findings (*Leach et al., 2021*), the number of MPO-positive cells was significantly higher in BC mice than in B mice; however, Fak loss did not further increase neutrophil infiltration in FBC mice (*Figure 4—figure supplement 1A*). Consistent with this, GSEA results showed that there was no difference in the expression of inflammatory response genes (*Liberzon et al., 2015*) in FBC mice and BC mice (*Figure 4—figure supplement 1B*). Together, these findings imply that Fak loss promotes tumor formation not by enhancing intestinal inflammation.

Next, we evaluated the roles of cellular senescence, apoptosis, cell proliferation, and *Lgr5* expression in cecal tumorigenesis in FBC mice. The results indicated that BRAF$^{V600E}$ was insufficient to trigger senescence evaluated by SA-β-galactosidase staining or apoptosis evaluated by the TUNEL staining in BC mice (*Figure 4—figure supplement 1C,D*). Bromodeoxyuridine (BrdU) incorporation assays confirmed mutant BRAF-induced hyperproliferation. However, Fak loss did not further enhance the BrdU incorporation rate (*Figure 4—figure supplement 1E*). These results indicated that *Ptk2* deletion promotes tumor formation not through modulating cellular senescence, apoptosis, and cell proliferation.

Given that BRAF$^{V600E}$ drives tumorigenesis through constitutive downstream ERK1/2 activation (*Wellbrock et al., 2004*), we examined the impact of Fak loss on ERK pathway transcriptional output.

GSEA analysis showed that ERK pathway output was significantly increased in BC mice (*Figure 4A*), which was consistent with the earlier report (*Pratilas et al., 2009*), but Fak loss did not further enhance it (*Figure 4F*). Wnt pathway activation (*Tong et al., 2021*) and activation of transcription co-factor YAP have been implied in BRAF$^{V600E}$-induced serrated tumorigenesis (*Leach et al., 2021*). In this study, our GSEA results also showed that the expression of intestinal Wnt signature genes (*Van der Flier et al., 2007*) and YAP target genes (*Wang et al., 2018*) were significantly higher in BC mice than in B mice (*Figure 2B,C*). Again, Fak loss did not further enhance the activations (*Figure 2G,H*). Together, these findings excluded the possibility that Fak loss promotes cecal tumor formation by enhancing ERK pathway output and activation of the Wnt and YAP pathways.

BRAF$^{V600E}$ poorly initiates colon cancer in mice due to oncogenic BRAF-induced tissue differentiation and loss of intestinal stem cells (*Tong et al., 2017*). With this, GSEA results showed increased expressions of intestinal differentiation signature genes (*Chong et al., 2009*; *Figure 4D*) and decreased expressions of intestinal stem cell signature genes (*Muñoz et al., 2012*; *Figure 4E*) in BC mice. Fak deletion did not reverse BRAF$^{V600E}$-induced tissue differentiation (*Figure 4I*) but significantly enhanced intestinal stemness (*Figure 4J*). These results revealed that Fak deletion promotes BRAF$^{V600E}$-induced cecal tumor formation through increasing intestinal stemness.

The adult stem cell marker Lgr5 and its relative Lgr4 are R-spondin receptors mediating R-spondin signaling and are critical for intestinal stemness (*Glinka et al., 2011*; *de Lau et al., 2011*). Mutant BRAF reduces *Lgr5* expression in the intestinal crypt (*Tong et al., 2017*; *Leach et al., 2021*). Our results confirmed the downregulation of *Lgr5* in the cecum crypt in BC mice, and we found that Fak loss did not restore *Lgr5* expression in FBC mice (*Figure 4—figure supplement 1F*). These results thus excluded the possibility that Lgr5 mediates Fak loss-induced intestinal stemness.

Prior studies show that the fetal type of intestinal stem cells has a strikingly different transcriptome than that of adult intestinal stem cells, and the receptor LGR4, but not LGR5, is essential for the cells (*Mustata et al., 2013*). In *Vil1-Cre*$^{ER}$;*Braf*$^{LSL-V600E/+}$;*Tgfbr1*$^{fl/fl}$ mice, the proximal colonic tumors exhibit fetal intestinal signature (*Leach et al., 2021*). Consistent with the notion that mutant *BRAF*-driven right-sided colonic tumors are fetal progenitor phenotypes, GSEA results confirmed enrichment of the fetal-type transcriptomic signatures (*Mustata et al., 2013*) in cecal mucosa in BC mice. The fetal signature was further enriched in FBC mice (*Figure 4K*). Accordingly, the immunoblotting analysis showed that the protein level of Lgr4 was increased in the intestine epithelium in FBC mice (*Figure 4I*). Consistent with the fact that intestinal Lgr5 expression was low in FBC mice (*Figure 4—figure supplement 1F*), FBC tumors mainly expressed Lgr4 but not Lgr5. In contrast, BC and *Apc*$^{min/+}$ tumors expressed both Lgr5 and Lgr4 (*Figure 4M*). These results suggest that upregulated Lgr4 mediated the intestinal stemness increase in FBC mice.

## FAK loss downregulates EGFR-dependent ERK phosphorylation to increase Lgr4 mRNA expression and protein stability

We addressed how Fak loss mediates Lgr4 increase. A prior study suggested that Wnt signaling maintains quiescent intestinal stem cell pools through suppression of the MAPK pathway in the intestine (*Kabiri et al., 2018*). Given the fact that Fak loss did not jeopardize ERK pathway transcriptional output (*Figure 4F*), Fak loss may increase intestinal stemness by inhibiting ERK phosphorylation. To test, we first compared the levels of phosphorylated ERK across the intestines in B mice, BC mice, and FBC mice. As anticipated, BRAF$^{V600E}$ increased p-ERK levels throughout the intestine (*Figure 5A*). FAK is positively involved in ERK1/2 activation (*Sulzmaier et al., 2014*). Consistent with this, in FBC mice, FAK deletion suppressed mutant BRAF-induced elevation of p-ERK (*Figure 5A*). The decoupling of ERK pathway output (no change) and the level of p-ERK (reduced) upon Fak loss is in line with a prior report suggesting that the level of ERK phosphorylation does not truthfully reflect ERK pathway activation (*Pratilas et al., 2009*).

We next examined how Fak loss altered BRAF$^{V600E}$-induced phosphorylation of ERK. A prior study found that FAK promotes EGFR signaling (*Sieg et al., 2000*), raising the possibility that FAK regulates ERK phosphorylation through EGFR. We then evaluated Egfr activation (represented by phosphorylated EGFR at tyrosine 1068) in the mice. The results showed that the level of phosphorylated Egfr$^{Y1068}$ was increased in BC mice throughout the intestine (*Figure 5B*). In FBC mice, Fak deletion moderately reduced BRAF$^{V600E}$-induced Egfr activation (*Figure 5B*) and suppressed Egfr downstream signal transduction as evidenced by the decreased levels of phosphorylated c-Raf$^{S338}$ and MEK1/2 $^{S217/221}$

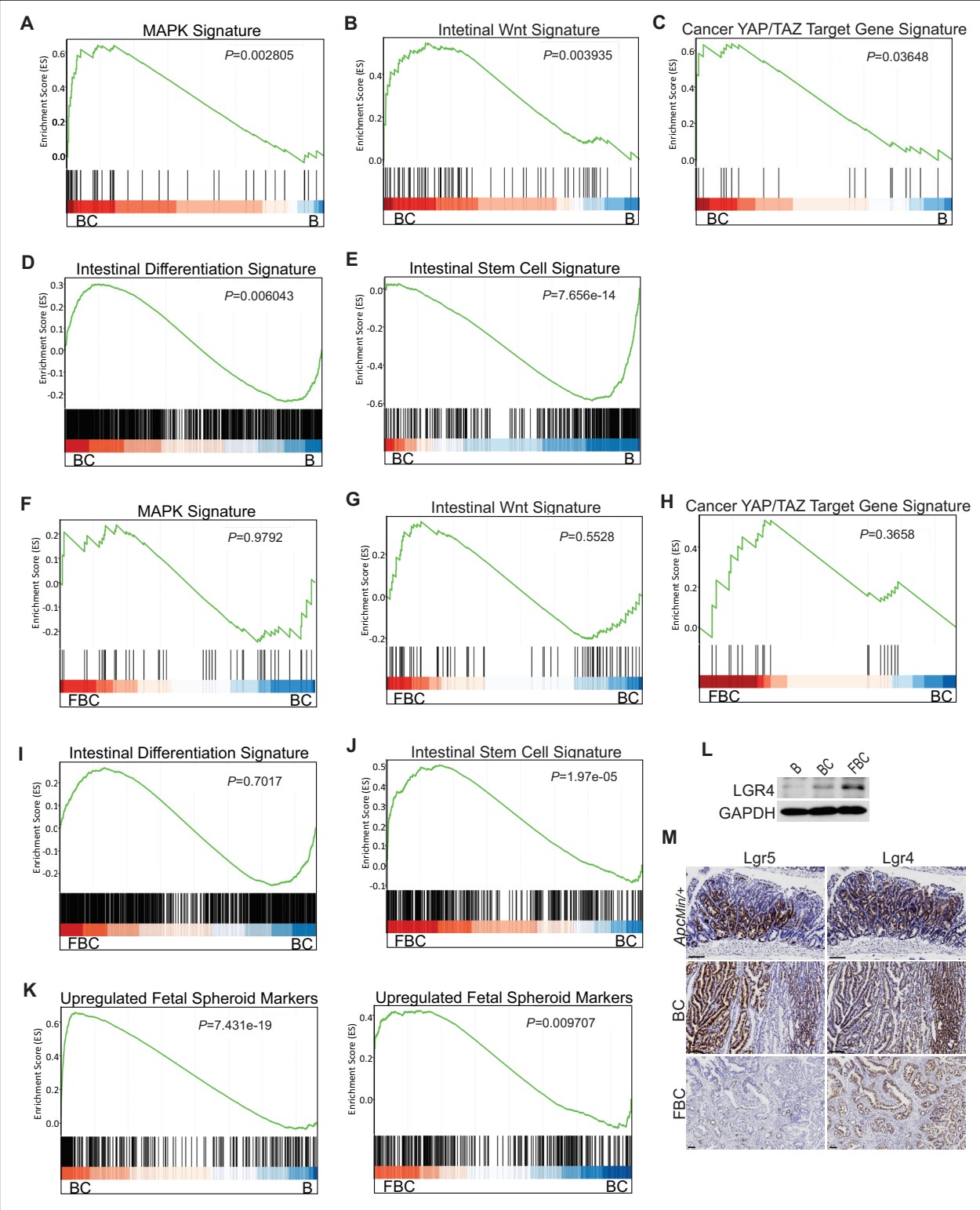

**Figure 4.** *BRAF*^V600E mutation and *Ptk2* loss-mediated changes in signaling pathways. GSEA analysis showing upregulation of MAPK signature (**A**), intestinal WNT signaling (**B**), YAP/TAZ target gene signature (**C**) and intestinal differentiation signature (**D**), and downregulation of intestinal stem cell signature (**E**) in the cecum of BC mice vs B mice (n=4 per group). GSEA plots revealed no significant change in MAPK signature (**F**), intestinal WNT signaling (**G**), YAP/TAZ target gene signature (**H**), and intestinal differentiation signature (**I**) in the cecum of FBC mice vs BC mice, but enrichment of stem cell signature in FBC mice (**J**) (n=4 per group). (**K**) GSEA analysis showing upregulation of upregulated fetal spheroid markers in the cecum of BC mice vs B mice, and further enrichment in the cecum of FBC mice vs BC mice (n=4 per group). (**L**) Immunoblotting analysis of LGR4 in the cecum from

*Figure 4 continued on next page*

*Figure 4 continued*

indicated 6-week-old mice. (**M**) Representative in situ hybridization (ISH) staining of tumor sections from *Apc^Min/+^*, BC, and FBC mice using *Lgr4* and *Lgr5* probes. Scale bars: 100 μm.

The online version of this article includes the following source data and figure supplement(s) for figure 4:

**Source data 1.** Uncropped and labelled gels for (*Figure 4*).

**Source data 2.** Raw unedited gels for (*Figure 4*).

**Figure supplement 1.** *Fak* loss has no impact on inflammation, senescence, apoptosis, proliferation, and *Lgr5* expression in the cecum in FBC mice.

in FBC mice (*Figure 5—figure supplement 1A*). To validate that EGFR indeed regulates BRAF^V600E^-induced ERK phosphorylation, we treated BC mice with the EGFR inhibitor erlotinib. Erlotinib treatment, without significantly reducing ERK pathway output (*Figure 5—figure supplement 1B*), indeed suppressed phosphorylation of C-RAF, MEK, and ERK (*Figure 5C*). Of note, Fak deletion had no impact on the level of p-EGFR and p-ERK in control mice (*Figure 5—figure supplement 1C*). Inhibition of Fak kinase activity by FAK inhibitor PF-562271 did not affect the phosphorylation of Egfr and ERK (*Figure 5D*), implying that the kinase activity of Fak is not involved in the FAK/EGFR/ERK regulation in BRAF^V600E^-induced serrated tumorigenesis.

FAK complexes with activated EGFR to promote EGFR signaling (*Sieg et al., 2000*). We assessed whether FAK interacts with EGFR in *BRAF^V600E^*-mutant cells. The results of co-immunoprecipitation using lysates from cecal mucosa confirmed the Fak-Egfr interaction and revealed that the Fak-Egfr interaction was increased in BC mice, and inhibition of Egfr appeared not to affect the Fak-Egfr binding (*Figure 5—figure supplement 1D*). ERK phosphorylation is refractory to EGFR inhibition in human BRAF^V600E^-mutant CRC cell lines (*Corcoran et al., 2012*; *Prahallad et al., 2012*); however, the FAK-EGFR interaction was still detected in HT29 CRC cells, and the interaction was not affected by either EGFR inhibition or FAK inhibition (*Figure 5—figure supplement 1E*). These results indicated that FAK/EGFR interaction alone is not sufficient for FAK to get involved in the regulation of MAPK signaling.

The contradictory results seen in BC mice and human *BRAF^V600E^*-mutant CRC cell lines could result from the differences between in vitro culture systems and in vivo. To test, we examined whether inhibition of Egfr leads to ERK inhibition in freshly isolated cecal crypts from BC mice and BC cecal organoids. The results showed that inhibition of Egfr did not reduce ERK phosphorylation, confirming that the contradictory findings resulted from in vitro and in vivo. We speculate that the lack of certain stromal factors in vitro is responsible for the EGFR's inability to transmit its signal to activate ERK.

Finally, we examined whether and how a reduction in ERK phosphorylation increases Lgr4 expression/stemness. Our results showed that treatment with MEK inhibitor increased the mRNA expression of LGR4 in human *BRAF^V600E^*-mutant CRC HT29 cells (*Figure 5F*) and BC mice (*Figure 5G*), uncovering a negative association between the level of ERK phosphorylation and mRNA expression of Lgr4. Of note, inhibition of ERK activation in BC mice was confirmed by the abrogation of ERK phosphorylation (*Figure 5G*) and suppression of ERK pathway transcriptional output (*Figure 5—figure supplement 2*). This negative association was further supported by our observation that the mRNA levels of Lgr4 were higher, albeit not statistically significant, in FBC mice than in BC mice (*Figure 5H*). Regulation of Lgr4 protein stability represents an important mechanism of modulating Lgr4 function (*Mancini et al., 2020*). Our cycloheximide chase analysis results showed that inhibition of ERK phosphorylation by MEK inhibitor treatment dramatically enhanced Lgr4 protein stability in *BRAF^V600E^*-mutant CRC cell line HT29 cells (*Figure 5I*). This finding revealed the inverse correlation between the level of ERK phosphorylation and the protein stability of Lgr4. These results suggest that Fak loss lowers BRAF^V600E^-induced ERK phosphorylation to increase Lgr4 mRNA expression and protein stability, thereby enhancing intestinal stemness and cecal tumor formation.

## Inhibition of ERK phosphorylation downregulates the level of E3 ubiquitin ligase NEDD4

We next investigated how the reduction of ERK phosphorylation increases Lgr4 stability. The HECT-domain E3 ligases NEDD4 (Neuronal precursor cell developmentally downregulated protein 4) and its homolog NEDD4L can ubiquitinate Lgr4, leading to its degradation (*Novellasdemunt et al., 2020*). Although the RNA-seq data showed no difference in mRNA expression levels of Nedd4 and Nedd4l

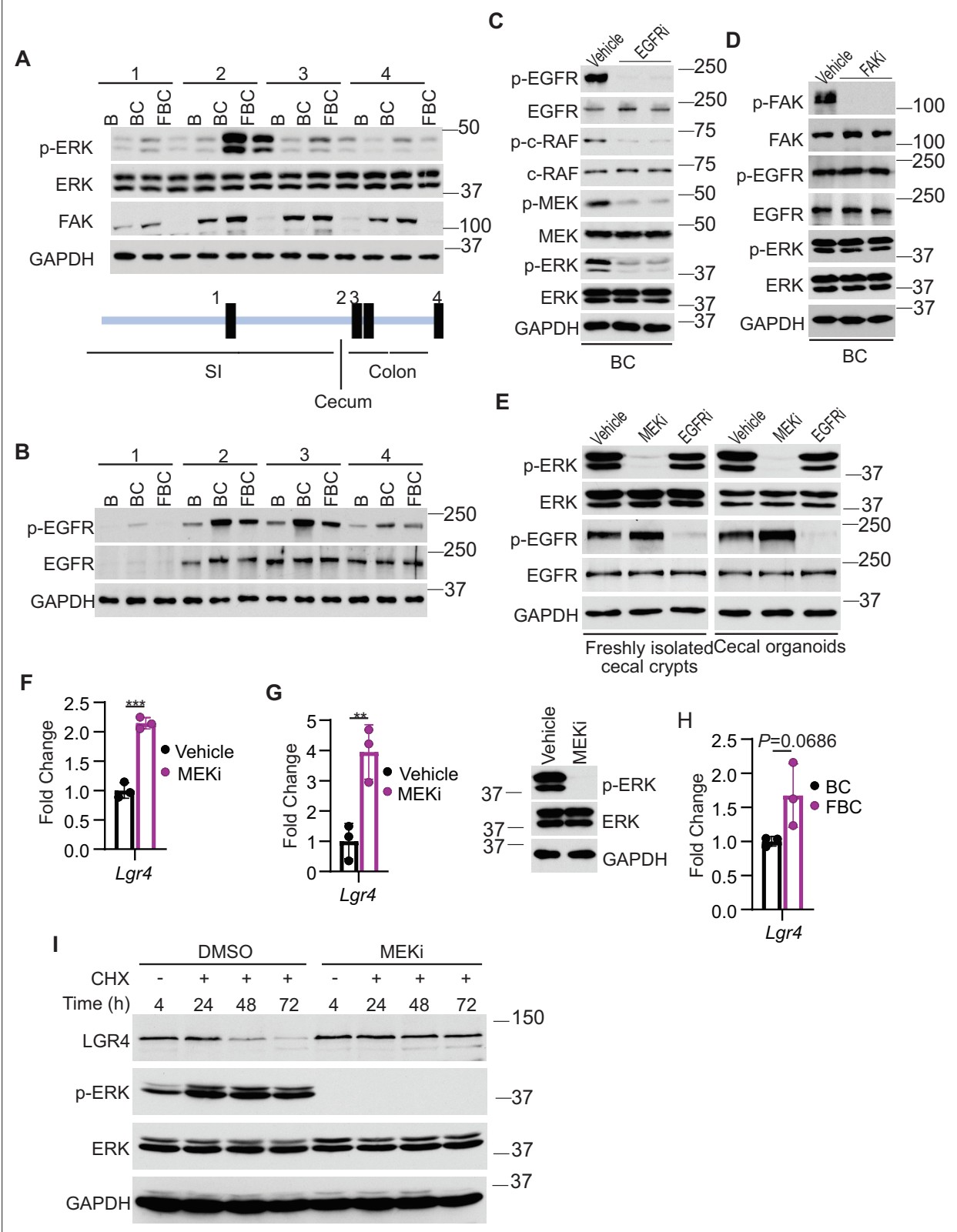

**Figure 5.** Fak loss inhibits ERK phosphorylation and upregulates Lgr4. (**A** and **B**) Immunoblotting analysis of intestinal mucosa lysates from indicated bowel subsites in indicated 6-week-old mice. (**C**) Immunoblotting analysis of cecum lysates from 6-week-old BC mice treated with vehicle or EGFR inhibitor erlotinib for 4 hr. Each lane represented a single mouse. (**D**) Immunoblotting analysis of cecum lysates from 6-week-old BC mice treated with vehicle or FAK inhibitor PF-562271 for 4 hr. Each lane represented a single mouse. (**E**) Immunoblotting analysis of lysates from freshly isolated cecal

*Figure 5 continued on next page*

*Figure 5 continued*

crypts and cecal organoids treated with DMSO, MEK inhibitor PD0325901, or erlotinib, respectively as described in Methods. (**F**) qRT-PCR of *Lgr4* using lysates from HT-29 cells treated with the vehicle and MEKi for 4 hr. Data presented as mean ± SD (***p<0.001; Student's *t*-test, two-tailed). (**G**) qRT-PCR of *Lgr4* using cecum lysates from BC mice treated with vehicle or MEKi for 6 hr. Data presented as mean ± SD (**p<0.01; Student's *t*-test, two-tailed). Abrogation of ERK phosphorylation at T202/Y204 in the cecum was confirmed by western blot. (**H**) qRT-PCR of *Lgr4* in cecum from BC and FBC mice (n=3 per group). Data presented as mean ± SD (p value calculated using two-tailed Student's *t*-test). (**I**) Immunoblotting analysis of the lysates from HT-29 cells treated with cycloheximide (100 μg/ml) and/or MEK inhibitor PD0325901 (10 μM) as indicated.

The online version of this article includes the following source data and figure supplement(s) for figure 5:

**Source data 1.** Uncropped and labelled gels for (*Figure 5*).

**Source data 2.** Raw unedited gels for (*Figure 5*).

**Figure supplement 1.** *Fak* loss downregulates *BRAF*^V600E-induced ERK phosphorylation.

**Figure supplement 1—source data 1.** Uncropped and labelled gels for (*Figure 5—figure supplement 1*).

**Figure supplement 1—source data 2.** Raw unedited gels for (*Figure 5—figure supplement 1*).

**Figure supplement 2.** qRT-PCR of selected ERK transcriptional output markers in cecum from vehicle- and MEKi-treated BC mice (n=3 per group).

in C57, BC, and FBC mice, the protein level of Nedd4, but not Nedd4l, was increased in BC mice then decreased in FBC mice (*Figure 6A*). To confirm that loss of ERK phosphorylation mediates the Nedd4 reduction, we treated the BC mice with MEK inhibitor and measured the protein levels of Nedd4 and Nedd4l. As shown in *Figure 6B*, MEK inhibitor treatment abrogated ERK phosphorylation and reduced the expression of Nedd4, accompanied by increased Lgr4 level. These data suggested

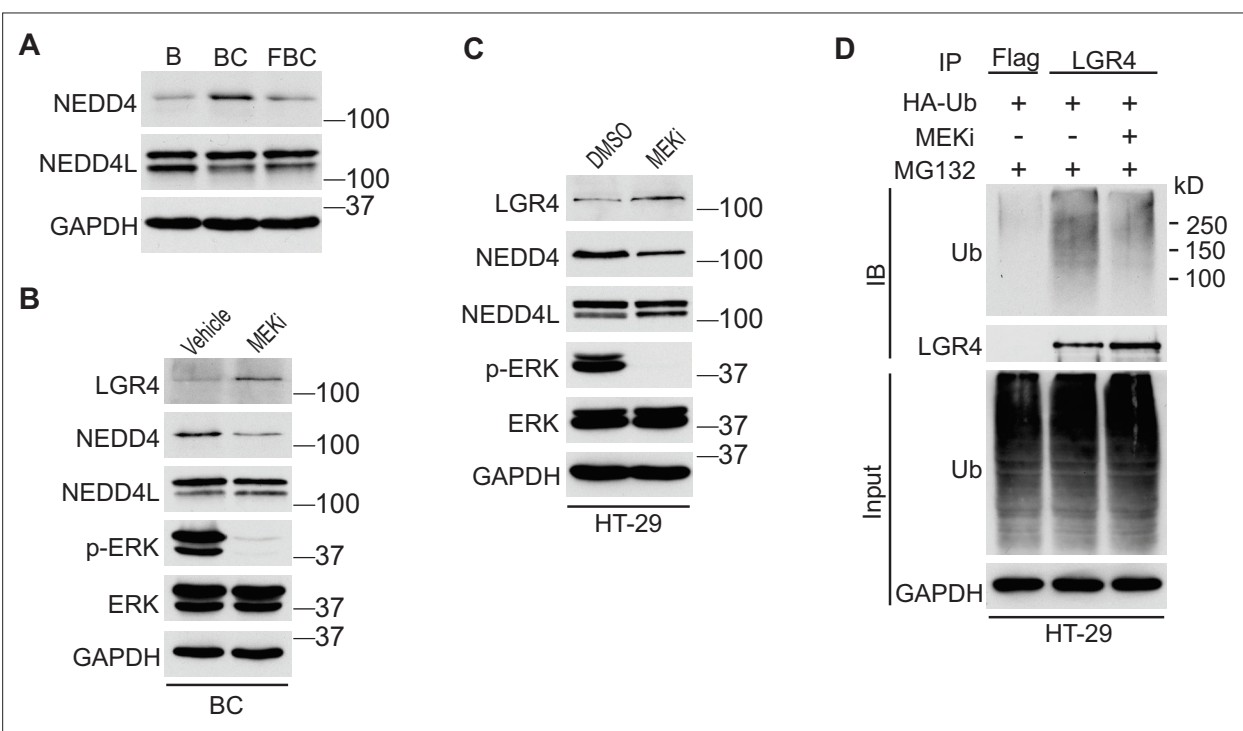

**Figure 6.** Inhibition of ERK phosphorylation stabilizes LGR4 through downregulating NEDD4. (**A**) Immunoblotting analysis of cecum lysates from indicated 6-week-old mice. (**B**) Immunoblotting analysis of cecum lysates from 6-week-old BC mice treated with vehicle or MEK inhibitor PD0325901. MEK inhibitor was given to the mice at a dose of 25 mg/kg three times at 12 hr intervals. Twenty-eight hours after the first treatment, the cecum mucosa was collected for immunoblotting. (**C**) Immunoblotting analysis of lysates from HT-29 cells treated with DMSO or 10 μM MEK inhibitor for 24 hr. (**D**) HT-29 cells were transfected with HA-Ubiquitin. One day later, the cells were treated with DMSO or 10 μM MEK inhibitor for 24 hr. Then all the cells were incubated with 10 μM MG132 for additional 4 hr. The cell lysates were collected for immunoprecipitation and immunoblotting with the indicated antibodies.

The online version of this article includes the following source data for figure 6:

**Source data 1.** Uncropped and labelled gels for (*Figure 6*).

**Source data 2.** Raw unedited gels for (*Figure 6*).

that reduced ERK phosphorylation reduces E3 ligase Nedd4 to increase Lgr4 stability. The decreased ubiquitination of LGR4 was confirmed in HT-29 cells. While treatment with MEK inhibitor inhibited the expression of NEDD4 (*Figure 6C*), it greatly reduced the ubiquitination of LGR4 (*Figure 6D*). Together, these data implied that reduction in ERK phosphorylation reduces the expression of E3 ubiquitin ligase Nedd4 in FBC mice to increase the Lgr4 level.

## FAK's influence on oncogenic MAPK-driven intestinal tumorigenesis depends on FAK's impact on ERK phosphorylation

Fak loss reduced ERK phosphorylation in FBC mice (*Figure 5A*) but not in control mice with wild-type *BRAF* (*Figure 5—figure supplement 1A*). To determine whether FAK is involved in other oncogenic MAPK-driven tumors, we generated *Vil1-Cre;Kras*^LSL-G12D/+^ (KC) mice and *Vil1-Cre;Kras*^LSL-G12D/+^;*Ptk2*^fl/fl^ (FKC) mice. In KC mice, the endogenous expression of oncogenic Kras induces serrated hyperplasia; however, high ERK activation-induced senescence prevents hyperplasia progression into dysplasia (*Bennecke et al., 2010*). As shown in *Figure 7A*, no tumor was found in KC mice (n=6, 9-months-old) and FKC mice (3-month-old, n=3; 6-month-old, n=3; 9-month-old, n=4). Immunoblotting results confirmed that Fak loss failed to influence the phosphorylation of Egfr or ERK (*Figure 7B*). The co-immunoprecipitation results showed that Fak complexed with Egfr in KC mice similarly as in BC mice (*Figure 7C*), implying that the noninvolvement of Fak was not due to the lack of Fak/Egfr interaction. A recent preprint (https://doi.org/10.1101/2020.07.02.185173) suggests that 'EGFR network oncogenesis cooperates with weak oncogenes in the MAPK pathway', which inspired us to propose the notion that EGFR participates in the regulation of ERK phosphorylation only when the p-ERK level is relatively low. In KC mice, KRAS^G12D^ induces extremely high levels of ERK phosphorylation, high enough to cause intestinal senescence (*Bennecke et al., 2010*). Given the level of increased p-ERK in KC mice, one would expect that ERK phosphorylation is EGFR-independent. The EGFR independence was confirmed by our results showing that pharmacologic abrogation of EGFR activation had no impact on KRAS^G12D^-induced ERK phosphorylation in KC mice (*Figure 7D*). Clinical findings further supported our notion. Anti-EGFR therapy is excluded for patients with *KRAS*-mutant CRC, supporting that EGFR has minimum impact on downstream MAPK signaling upon *KRAS* mutation. However, when ERK activation is inhibited by KRAS^G12C^ inhibitors, EGFR signaling acts as the dominant mechanism of colorectal cancer resistance to KRAS^G12C^ inhibitors (*Amodio et al., 2020*).

To address whether FAK downregulation is specific to human *BRAF*-mutant CRCs, we compared FAK expression levels in CRCs with different driver mutations using the TCGA database. TCGA analysis revealed that *PTK2* mRNA levels were significantly lower in *BRAF*-mutated CRCs than in *APC*-mutated CRCs or *KRAS*-mutant CRCs (*Figure 7E*). This result is consistent with the result seen in mice, again, it suggests that FAK is not involved in the regulation of *KRAS*-mutant CRCs.

In mice, mutant BRAF-induced ERK activation is cancer stage-dependent with significantly higher levels of phosphorylated ERK in high-grade dysplasia and carcinoma (*Rad et al., 2013*), suggesting that different tumor stages may require different levels of p-ERK. If FAK is a key regulator of ERK phosphorylation in mutant *BRAF*-induced serrated tumorigenesis in patients, one would expect the level of FAK may increase as the tumors progress. Consistent with this notion, we observed that FAK levels were higher in BRAF-mutant CRCs than in BRAF-mutant polyps (*Figure 1A*), TCGA analysis (*Figure 7E*) further confirmed that FAK expression was restored to a level similar to normal intestines, albeit still significantly lower than in APC mutant or KRAS mutant CRCs (*Figure 7E*).

In patients, BRAF mutations are divided into two groups: Activator and amplifier mutation (*Yaeger and Corcoran, 2019*). In CRC, the majority (80%–90%) of activating mutations in BRAF are V600E (*Rajagopalan et al., 2002*). Among these mutants, based on their kinase activities, BRAF^V600E^ belongs to the high-activity mutants, and the rest of the mutants except G595R (with impaired BRAF kinase activity in vitro but still induce constitutive ERK activation in vivo) are intermediate activity mutants (*Wan et al., 2004*). If mutant BRAF-induced ERK phosphorylation needs to reach a 'just-right' level via FAK downregulation in patients, one would expect that the degree of FAK downregulation is BRAF mutant activity-dependent, and there could be a correlation between the activity of BRAF mutants and the degree of FAK reduction. Consistent with this speculation, TCGA data analysis confirmed that CRCs with *BRAF*^V600E^ mutation had lower FAK expression than CRCs with non-V600E mutations and *BRAF* wild-type CRCs (*Figure 7F*). Although the differences between V600E and non-V600E groups were not statistically significant due to limited sample numbers, they might be biologically relevant.

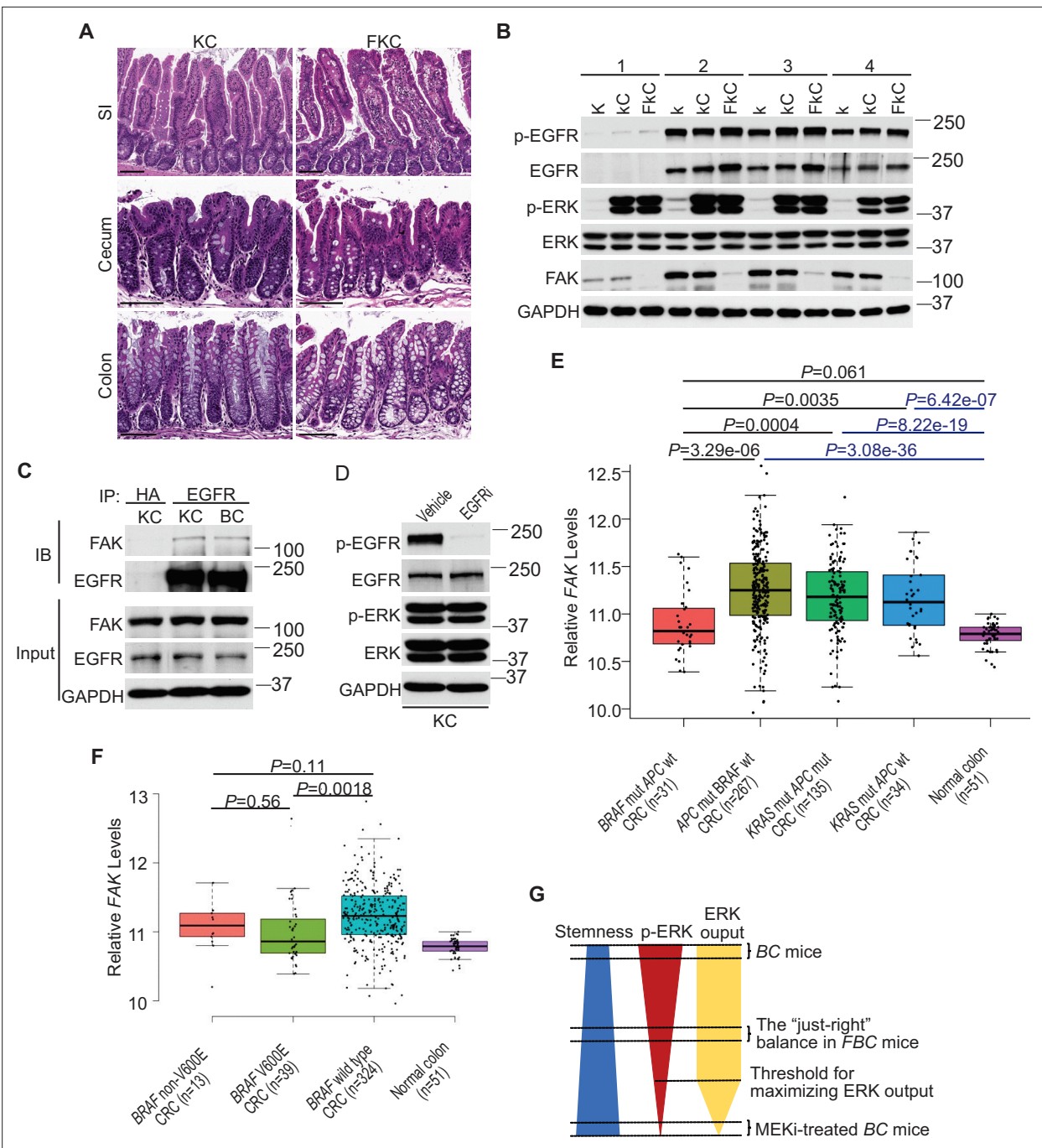

**Figure 7.** ERK activation is FAK/EGFR-independent in KC mice. (**A**) Representative hematoxylin and eosin (H&E) staining of the small intestine, cecum, and colon from indicated 9-month-old mice. (**B**) Immunoblotting analysis of intestinal mucosa lysates from indicated bowel subsites in indicated 6-week-old mice. (**C**) The cecal mucosa lysates from 6-week-old KC and BC mice were used for immunoprecipitation and immunoblotting with the indicated antibodies. (**D**) Immunoblotting analysis of cecum lysates from 6-week-old KC mice treated with vehicle or EGFR inhibitor erlotinib for 4 hr. (**E** and **F**) Comparison of *FAK* expression levels between CRCs with indicated mutations by analysis of TCGA RNA-sequencing dataset. Data were analyzed for statistical significance using a Student t-test. (**G**) Diagram of the 'just-right' MAPK signaling model in the serrated pathway.

The online version of this article includes the following source data for figure 7:

**Source data 1.** Uncropped and labelled gels for (*Figure 7*).

**Source data 2.** Raw unedited gels for (*Figure 7*).

## Discussion

The current study finds that in *BRAF*<sup>V600E</sup>-mutant intestinal epithelium, elevating the p-ERK level to a minimum threshold is sufficient to maximize the pathway transcriptional output, that is, only lowering the p-ERK level below the threshold will significantly abrogate the ERK pathway transcriptional output. Due to the negative association between ERK phosphorylation and intestinal stemness, any increase in ERK phosphorylation will decrease intestinal stemness (*Figure 6G*). In *BRAF*<sup>V600E</sup>-mutant intestinal epithelium, ERK phosphorylation is EGFR/RAS/c-RAF-dependent. The involvement of EGFR provides an opportunity for non-MAPK pathway factors such as FAK to participate in the regulation of ERK phosphorylation to influence the biological outcomes of *BRAF* mutation. This study has established the first 'just-right' MAPK signaling model of BRAF<sup>V600E</sup>-induced tumor formation (*Figure 7G*). Our results show that by lowering BRAF<sup>V600E</sup>-induced ERK phosphorylation, Fak loss, without jeopardizing the ERK pathway transcriptional output, enhances mRNA expression and protein stability of Lgr4, thereby increasing intestinal stemness and promoting cecal tumor formation in mice.

High-level activation of oncogenes (e.g. KRAS, BRAF, and c-MYC) triggers intrinsic tumor suppression (*Bennecke et al., 2010*; *Michaloglou et al., 2005*; *Dankort et al., 2007*; *Sarkisian et al., 2007*; *Murphy et al., 2008*). Genetic abrogation of tumor suppressors such as p53 or p16 revokes the tumor-suppressive barrier, thereby facilitating oncogene-induced tumorigenesis (*Carragher et al., 2010*; *Bennecke et al., 2010*; *Dankort et al., 2007*; *Sarkisian et al., 2007*). Cooperation with other oncogenic stimulation, such as co-expression of c-MYC and KRAS, ultraviolet radiation on melanocytes expressing BRAF<sup>V600E</sup>, can also break the suppressive barrier (*Land et al., 1983*; *Viros et al., 2014*). In cellular models (*Kidger et al., 2017*; *Unni et al., 2018*), overexpression of MKP/DUSPs evades high ERK activation-induced tumor suppression. Whether and how the suppressive barrier can be avoided or reduced in vivo has never been experimentally tested. The current study is the first demonstration that mutant BRAF-induced activation of ERK signaling is tuneable in vivo, and by tuning ERK activation to alter the suppressive barrier, FAK regulates BRAF transforming activity.

In *BRAF*-mutated melanoma, a complete shutdown of the MAPK pathway is necessary for significant tumor response (*Bollag et al., 2010*). In patients with *BRAF*<sup>V600E</sup>-mutated CRCs, a combination of encorafenib, cetuximab, and binimetinib (MEK inhibitor) treatment increased the response rate to 26% (*Kopetz et al., 2019*), highlighting the importance of complete ERK pathway inhibition. However, the inverse correlation between the level of phosphorylated ERK and the level of stemness/Lgr4 expression seen in mutant BRAF-expressing intestinal epithelial cells let us speculate that inhibition of ERK phosphorylation may cause stemness increases in *BRAF*-mutated CRC cells. The molecular mechanisms underlying ERK phosphorylation inhibition-mediated stemness increase remain to be determined. Given the importance of cancer cell stemness in treatment resistance (*Batlle and Clevers, 2017*), we propose that the optimal treatment outcome can only be achieved when the inhibition of ERK phosphorylation-mediated stemness increase is simultaneously suppressed.

In sum, the current study reveals the existence of a balance—between the level of phosphorylated ERK, the level of ERK pathway output, and the level of intestinal stemness. Our results show that the 'just-right' balance optimal for *BRAF*<sup>V600E</sup>-induced cecal tumor formation can be achieved through FAK alteration. Achieving optimal treatment response in *BRAF*-mutated CRC patients, though, may require abrogation of the p-ERK-stemness regulatory link. That said, the current study could have profound implications for the development of new anticancer agents and new treatment approaches for patients with *BRAF*-mutated CRC.

## Methods

### Mice and treatment

All animal procedures were performed according to protocols approved by the Institutional Animal Care and Use Committee at the University of Pittsburgh. Mice were fed a standard diet (diet ID 5P75; Purina LabDiet, St. Louis, MO). *Ptk2*<sup>fl/fl</sup> mice were received from the Mutant Mouse Resource & Research Centers (MMRRC, cat. no. 009967-UCD). *Villin-Cre* (cat. no. 021504), *Braf*<sup>LSL-V600E/+</sup> (cat. no. 017837), *Kras*<sup>LSL-G12D/+</sup> (cat. no. 008179) and *Rosa26*<sup>LSL-tdTomato</sup> (cat. no. 007914) mice were obtained from the Jackson Laboratory. Genotyping was performed according to the protocols provided by MMRRC and the Jackson Laboratory. *Villin-Cre* and *Braf*<sup>LSL-V600E/+</sup> mice were crossed to get the BC mice. The littermates harboring *Braf*<sup>LSL-V600E</sup> allele were used as controls whenever available. To get the FBC mice,

*Ptk2*<sup>fl/fl</sup> mice were first crossed with *Villin-Cre* mice and *Braf*<sup>LSL-V600E/+</sup> mice, respectively. The offspring *Villin-Cre;Ptk2*<sup>fl/+</sup> and *Braf*<sup>LSL-V600E/+</sup>;*Ptk2*<sup>fl/+</sup> mice were further crossed with *Ptk2*<sup>fl/fl</sup> mice to get the *Villin-Cre;Ptk2*<sup>fl/fl</sup> (FC) and *Braf*<sup>LSL-V600E/+</sup>;*Ptk2*<sup>fl/fl</sup> (FB) mice. The FBC mice were finally obtained by crossing FC and FB mice. The same strategy was used to generate the FKC mice. BC, FBC, KC and FKC mice were euthanized at the indicated age to evaluate the tumor formation. *Villin-Cre* mice and *Rosa26*<sup>LSL-tdTomato/LSL-tdTomato</sup> mice were crossed to get the *Villin-Cre; Rosa26*<sup>LSL-tdTomato/+</sup> mice.

For Bromodeoxyuridine (BrdU) labeling, 6-week-old mice were given BrdU (MilliporeSigma) at a dose of 100 mg/kg by intraperitoneal injection two hours prior to harvesting. For inhibitor treatment, six-week-old mice were given vehicle (a mixture of 50% DMSO and 50% PEG 400), PF-562271 (60 mg/kg in vehicle) or Erlotinib (100 mg/kg in the vehicle) by a single oral gavage 4 hr (for immunoblotting) or 6 hr (for qRT-PCR analysis of ERK output genes) before harvesting. MEK inhibitor PD0325901 was given to mice by oral gavage at a dose of 25 mg/kg in the vehicle. All experiments were performed in both male and female mice.

## Plasmid and transient transfection

pcDNA3-HA-Ubiquitin (18712) was from Addgene. Plasmid transient transfections were performed using PolyJet In Vitro DNA Transfection Reagent (SignaGen) according to the manufacturer's instructions.

## Cell culture and treatment

HT-29 cells were obtained from the American Type Culture Collection (ATCC) and cultured in DMEM supplemented with 5% fetal bovine serum, 100 units/ml penicillin and 100 μg/ml streptomycin, in a 37 °C humidified incubator containing 5% CO2. To study the interaction between FAK and EGFR in HT-29 cells, the cells were treated with DMSO, PF-562271 (5 μM) or erlotinib (10 μM) for 1 hr before harvested for immunoprecipitation. To study the ubiquitination of LGR4, HT-29 cells were treated with DMSO or 10 μM MEK inhibitor PD0325901 for 24 hr. Then 10 μM MG132 was added to the culture medium and incubated for additional 4 hr before harvesting the cells for immunoprecipitation.

## Protein stability assay

HT-29 cells were seeded twenty-four hours before the experiments. The cells were treated with 100 μg/ml cycloheximide (Selleck Chemicals), 10 μM MEK inhibitor PD0325901, or their combination as indicated. Then the cells were harvested, and the whole cell lysates were used for immunoblotting.

## Organoid culture and treatment

Mouse organoids were isolated according to the published protocol with some modifications (*Sugimoto and Sato, 2017*). Briefly, the cecum of the BC mouse was rinsed with cold PBS, cut into small pieces, and washed eight times in cold PBS by gently pipetting. The fragments were incubated in 10 mM EDTA diluted in PBS for 8 min in a 37 °C tube rocker. Then the EDTA solution was removed and the tissue was pipetted 10 times in cold PBS. The supernatant was collected and centrifuged at 300 × *g* for 3 min at 4 °C. The cell pellet was washed with DMEM/F-12 medium and centrifuged at 400 × *g* for 3 min at 4 °C. The pellet was resuspended in Cultrex Reduced Growth Factor Basement Membrane Extract, Type R1 (R&D Systems), and seeded into a 24-well plate. Organoids were cultured using Mouse IntestiCult Organoid Growth Medium (STEMCELL Technologies) in a 37 °C humidified incubator containing 5% CO$_2$. The medium was changed every other day. For inhibitor experiments, the freshly isolated crypts (one hour after seeding) and organoids (five days after seeding) were treated with 10 μM EGFR inhibitor erlotinib and 10 μM MEK inhibitor PD0325901, respectively, for two hours. To isolate protein for immunoblotting after treatment, the crypt cultures were scraped and suspended in 500 μl of TrypLE Express containing 10 μM EGFR inhibitor or 10 μM MEK inhibitor and incubated at a 37 °C water bath for 5 min with occasional agitation. After the addition of 500 μl of DMEM/F-12 medium, the crypt cultures were centrifuged at 400 × *g* for 3 min at 4 °C. The cell pellets were resuspended in cold PBS and centrifuged again. The final pellets were lysed in RIPA buffer (Alfa Aesar) supplemented with protease inhibitor and phosphatase inhibitor (Thermo Fisher Scientific). Crypt cultures treated with DMSO were used as controls. The lysates were quantified and resolved by sodium dodecyl sulfate-polyacrylamide gel electrophoresis (SDS-PAGE) and blotted with the indicated antibodies.

## Immunoblotting and immunoprecipitation

After the mice were euthanized, the entire intestines were immediately removed and rinsed twice with ice-cold PBS. The mucosal layers of the small intestine (about 1 cm length), colon (about 1 cm length), and cecum (entire cecum, without appendix) were harvested by scraping with a blade and all procedures were performed on ice. The freshly collected tissue was lysed in RIPA buffer supplemented with protease inhibitor and phosphatase inhibitor. The lysates were quantified and resolved by SDS-PAGE and blotted with the indicated antibodies. SuperSignal Western Blot Enhancer (Thermo Fisher Scientific) was used to enhance the blotting signal when needed. To detect the interaction between FAK and EGFR, the tissue lysates were pre-cleared with Protein G-sepharose beads at 4 °C for 30 min. The cleared lysates were incubated with anti-EGFR antibody conjugated to agarose (Santa Cruz Biotechnology) or anti-HA affinity gel (MilloporeSigma) at 4 °C for 4 hr. The immunoprecipitates were washed three times with lysis buffer containing 20 mM Tris-HCl, pH 7.5, 150 mM NaCl, 1 mM EDTA, 1% NP40, and 10% Glycerol, and subjected to SDS-PAGE followed by immunoblotting. The same protocol was used for immunoprecipitation experiments with HT-29 cell lysates. The cell lysates precipitated with anti-HA or anti-Flag beads were used as controls. The antibodies used for immunoblotting are shown in (*Supplementary file 2*). All experiments were independently repeated at least three times.

## Immunohistochemistry, in situ hybridization, BrdU staining, TUNEL staining, and histopathology

The de-identified human colon tissue samples from *BRAF*^V600E-mutated CRC patients were provided by the University of Pittsburgh School of Medicine, Department of Pathology tissue core. For mouse tissue sections, the mouse intestine was dissected out, rinsed twice with ice-cold PBS, fixed overnight in 10% neutral buffered formalin at 4 °C, embedded in paraffin, and finally cut into 5 µm sections. The sections were deparaffinized in xylenes and rehydrated in graded alcohol solutions, followed by washes in distilled water. Antigen retrieval was performed for 15 min in boiling pH 8 EDTA buffer (Abcam). The sections were allowed to cool to room temperature and then washed with PBS. The endogenous peroxidase was blocked with 3% hydrogen peroxide for 10 min. After washing with PBS, the sections were blocked with 20% goat serum diluted in PBS for 45 min. Sections were then incubated overnight at 4 °C in a humidified chamber with primary antibodies diluted in 3% BSA. Primary antibodies used in this study are listed in (*Supplementary file 2*). The sections were washed with PBS and incubated with secondary antibodies for 1 hr at room temperature. Color visualization was performed with 3.3'-diaminobenzidine until the brown color fully developed. The sections were counterstained with hematoxylin, dehydrated, and coverslipped with permanent mounting media. The slides were scanned using the Aperio digital pathology slide scanner (Leica Biosystems). The images were analyzed using Aperio ImageScope software.

In situ hybridization (ISH) was performed using the RNAscope 2.5 HD Reagent Kit-BROWN (Advanced Cell Diagnostics) according to the manufacturer's instructions. The following probes from Advanced Cell Diagnostics were used: *Lgr5* (cat. no. 312171) and *Lgr4* (cat. no. 318321).

BrdU staining was performed on formalin-fixed paraffin-embedded (FFPE) tissue sections using a monoclonal anti-BrdU antibody (MilloporeSigma) as described by the manufacturer. For Terminal deoxynucleotidyl transferase dUTP nick-end labelling (TUNEL) staining, the FFPE tissue sections were deparaffinized, treated with proteinase K and labeled using the In Site Cell Death Death Detection Kit POD (MilloporeSigma) according to the manufacturer's instructions. To quantify the results of BrdU, TUNEL and RFP staining, thirty crypts/villi per mouse were scored for three mice in each group.

Myeloperoxidase (MPO) was used as the marker for neutrophils. Ten random-chosen 500 µm-length cecum sections were evaluated for each mouse. MPO^+ cells within the band of lamina propria, immediately beneath and surrounding the crypts, were counted. Three mice in each group were analyzed. H&E-stained intestinal sections were evaluated for tumor stage by a board-certified GI pathologist (Dr. SF Kuan).

## Quantitative reverse-transcription PCR analysis

Total RNA was extracted from the mucosal layer of the mouse intestine or HT-29 cells using the RNeasy Mini Kit (QIAGEN). The DNase-treated RNA was reverse-transcribed using SuperScript III reverse transcriptase (Invitrogen). The PCR reactions were performed on the CFX Connect Real-Time PCR Detection System (Bio-Rad Laboratories) using SsoAdvanced Universal SYBR Green Supermix

(Bio-Rad Laboratories). The PCR thermal cycle conditions were as follows: denature at 95 °C for 30 s and 40 cycles for 95 °C, 10 s; 60 °C, 30 s. The specificity of the PCR products was determined by the melting curve analysis. *β-actin* was selected as an internal reference gene. The sequences of PCR primers are shown in (*Supplementary file 3*).

## Senescence-associated (SA) β-galactosidase staining

After the mice were euthanized, the cecum was immediately removed and rinsed with ice-cold PBS. The tissues were frozen in dry ice after the excess liquid was carefully removed using filter paper. Then the tissues were embedded in OCT compound and cut into 10 μm sections. The assays were performed using the Senescence β-Galactosidase Staining Kit (Cell Signaling Technology) according to the manufacturer's instructions. The sections were counterstained with hematoxylin before being dehydrated and coverslipped with mounting media.

## MSI analysis

The DNA was extracted from FFPE tissue sections using QIAamp DNA FFPE Tissue Kit (Qiagen). Cecal hyperplasia samples were from 6-week-old FBC mice. Cecal tumor samples were from 9- to 14.5-month-old FBC mice. Cecal tissue of 6-week-old B mice was used as control. According to a prior report (*Nakanishi et al., 2018*), five microsatellite repeat markers, Bat24, Bat26, Bat30, Bat37, and Bat64, were used for MSI analysis. PCR amplification was carried out in a multiplex reaction using HSTaq polymerase (Takara Bio, Japan), with primer concentrations 0.5 μM. The thermal cycling conditions were as follows: initial denaturation at 95 °C for 5 min; followed by 35 cycles of 95 °C for 30 s, 60 °C for 30 s, and 72 °C for 30 s; then a final extension step at 68 °C for 30 min. PCR fragments were analyzed by capillary electrophoresis, ABI3130XL (Life Technologies), and the GeneMapper ID3.2 program (Life Technologies). Tumor samples with greater or equal 40% MSI were classified as MSI-high (MSI-H), less than 40% as MSI-low (MSI-L), and samples without alterations were classified as MSS.

## RNA-seq and data analysis

Total RNA was extracted from the cecal tissues of indicated mice using the RNeasy Mini Kit (QIAGEN). After DNase I treatment and performing quality control (QC), 200 ng of high-quality total RNA was proceeded to library construction. Oligo(dT) magnetic beads were used to isolate mRNA. The mRNA was fragmented randomly by adding fragmentation buffer, then the cDNA was synthesized using mRNA template and random hexamers primer. Short fragments are purified and resolved with EB buffer for end repair and single nucleotide A (adenine) addition. After that, the short fragments were connected to sequencing adapters. The double-stranded cDNA library was completed through size selection and PCR enrichment. Agilent 2100 Bioanaylzer and ABI StepOnePlus Real-Time PCR System were used in the quantification and qualification of the sample library. Finally, the qualified RNA-seq libraries were sequenced using Illumina NovaSeq6000 in CD Genomics (Shirley, NY) after pooling according to its effective concentration and expected data volume. The FastQC tool was used to perform basic statistics on the quality of the raw reads. Sequencing adapters and low-quality data were removed by Cutadapt (version 1.17). The alignment tool Salmon (version 0.13.1) was employed to quantify transcript expression based on mm10 reference genome. Output files from Salmon were imported into R (V.4.2.0) and analyzed by DESeq2 package (V1.36.0) to identify differentially expressed genes. All genes were ranked by log2(fold change) and used to check the gene set enrichment by using clusterProfiler *Carragher et al., 2010* (V.4.4.1) in R. The following gene sets were used: MAPK signature *Pratilas et al., 2009*; intestinal Wnt signature *Van der Flier et al., 2007*; cancer YAP/TAZ target gene signature *Wang et al., 2018*; intestinal differentiation signature *Chong et al., 2009*; intestinal stem cell signature *Muñoz et al., 2012*; the Hallmark Inflammatory Response gene set (Broad Institute) *Liberzon et al., 2015*; upregulated fetal spheroid markers *Mustata et al., 2013*; upregulated and downregulated genes in human SSA/P *Kanth et al., 2016* (only genes in human SSA/Ps with fold increase >2 or fold decrease<-2 with FDR <0.05 were used).

## Whole exome sequencing

DNA was extracted from the cecal tumor of 12-month-old FBC mice using DNeasy Blood & Tissue Kits (Qiagen). Sequencing libraries were generated using Agilent SureSelect mouse All Exon Kit

(Agilent Technologies) following the manufacturer's instructions and index codes were added to attribute sequences to each sample. DNA samples were sonicated using a hydrodynamic shearing system (Covaris) to generate 180–280 bp fragments. The remaining DNA overhangs were converted into blunt ends by exonuclease/polymerase. After the adenylation of 3' ends, DNA fragments were ligated with adapter oligonucleotides. The fragments with adapters on both ends were selectively enriched using PCR. Then the library was hybridized in the liquid phase with biotin-labeled probes, followed by the capture of the exons using streptomycin-coated magnetic beads. Captured libraries were enriched by PCR to add index tags to prepare for hybridization. The resulting products were then purified using the AMPure XP System (Beckman Coulter) and quantified using the Agilent High Sensitivity DNA Assay on the Agilent Bioanalyzer 2100 System. The qualified libraries were sequenced using Illumina NovaSeq6000 in CD Genomics (Shirley, NY) after pooling according to its effective concentration and expected data volume. For the alignment step, BWA is utilized to perform reference genome alignment with the reads contained in paired FASTQ files. For the first post-alignment processing step, Picard tools are utilized to identify and mark duplicate reads from BAM file. The variant calling was performed by using GATK HaplotypeCaller.

## Analysis of CRC patient data

TCGA RNA-seq data and mutation data of all cancer types were collected from Xena database (https://xenabrowser.net/datapages/), i.e., TCGA Pan-Cancer (PANCAN), which includes 376 CRC tumor samples and 51 matched normal samples. Expression data for *PTK2* and mutation data for *BRAF* were extracted for analysis. The difference between the two groups was evaluated using the Student *t*-test (two-tailed, pairwise).

# Additional information

## Funding

| Funder | Grant reference number | Author |
| --- | --- | --- |
| National Cancer Institute | CA236965 | Jing Hu |
| National Institute of Diabetes and Digestive and Kidney Diseases | DK120698 | Farzad Esni |
| National Institute of Allergy and Infectious Diseases | AI158824 | Farzad Esni |

The funders had no role in study design, data collection and interpretation, or the decision to submit the work for publication.

## Author contributions

Chenxi Gao, Data curation, Formal analysis, Validation, Investigation, Visualization, Methodology; Huaibin Ge, Software, Formal analysis; Shih-Fan Kuan, Resources, Formal analysis; Chunhui Cai, Xinghua Lu, Software; Farzad Esni, Robert E Schoen, Jing H Wang, Investigation; Edward Chu, Conceptualization, Investigation; Jing Hu, Conceptualization, Data curation, Formal analysis, Supervision, Funding acquisition, Investigation, Methodology, Writing - original draft, Writing - review and editing

## Author ORCIDs

Xinghua Lu http://orcid.org/0000-0002-8599-2269
Farzad Esni http://orcid.org/0000-0002-0342-6862
Jing Hu http://orcid.org/0000-0002-1849-4550

## Ethics

This study was performed in strict accordance with the recommendations in the Guide for the Care and Use of Laboratory Animals of the National Institutes of Health. All of the animals were handled according to approved institutional animal care and use committee (IACUC) protocols (#21110173) of the University of Pittsburgh. The protocol was approved by the Committee on the Ethics of Animal

Experiments of the University of Pittsburgh (PHS Assurance Number: D16-00118). All surgery was performed under sodium pentobarbital anesthesia, and every effort was made to minimize suffering.

Reviewer #1 (Public Review): https://doi.org/10.7554/eLife.94605.2.sa1
Reviewer #2 (Public Review): https://doi.org/10.7554/eLife.94605.2.sa2
Reviewer #3 (Public Review): https://doi.org/10.7554/eLife.94605.2.sa3
Author response https://doi.org/10.7554/eLife.94605.2.sa4

---

## Additional files

### Supplementary files
• Supplementary file 1. The results of whole-exome sequencing on paired tumors (n=2) and neighboring mucosa show no additional driver mutations were detected in the cecal tumors.
• Supplementary file 2. List of the antibodies used in this study.
• Supplementary file 3. List of the PCR primers used in this study.
• MDAR checklist

### Data availability
Sequencing data have been deposited in GEO under accession codes GSE266355.

The following dataset was generated:

| Author(s) | Year | Dataset title | Dataset URL | Database and Identifier |
|---|---|---|---|---|
| Chenxi G, Huaibin H | 2024 | Gene expression profile of the intestines at the different locations | https://www.ncbi.nlm.nih.gov/geo/query/acc.cgi?acc=GSE266355 | NCBI Gene Expression Omnibus, GSE266355 |

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
