## [Editor Report · eLife assessment]

In this **important** study, the authors use a genetically engineered mouse model to reveal a tumor suppressive role for focal adhesion kinase in right-sided colon cancer. The evidence in support of the authors' claims is generally **solid**, although the data supporting the mechanism through which FAK deletion promotes tumorigenesis are **incomplete**. This work will be of interest to cancer researchers and others studying the biological consequences of tuning signal transduction pathways.

---

## [Referee Report · Reviewer #1 (Public Review)]

Summary:

The authors provide solid evidence with a mouse model as well as supporting in vitro and analysis of clinical samples that loss of Fak increases the development of BRAF V600E-induced dysplastic lesions and carcinomas in the cecum via downregulation of Egfr-mediated Erk phosphorylation. This fine-tuning of Erk phosphorylation increases the expression of Lrg4 mRNA expression and promotes Lrg4 stability through downregulation of the E3 ubiquitin ligase Nedd4. The high Lrg4 expression correlates with an increased intestinal stem cell transcriptional signature that the authors suggest drives higher rates of transformation. This provides important insight that factors such as FAK may be able to modulate MAPK-driven tumorigenesis in specific circumstances. The data presented here are largely specific to the cecum. While these specific findings may ultimately have practical implications for human CRC outside the cecum and even therapeutic implications, these remain unexplored and will be a point for future investigations.

Strengths:

The authors use a mouse model (intestinal specific BRAF V600E +/- Fak knockout) as well as supporting in vitro analyses and clinical sample characterization to support their model. For both in vitro and in vivo studies, the authors use a combination of genetic and pharmacologic (including EGFR, FAK, and MEK inhibitors) tools to modulate the MAPK pathway. They also use a combination of transcriptional (RNA-Seq) and protein (IHC and Western blotting) readouts to support their proposed model. Importantly, they use a distinct mouse model (mutant Kras) to demonstrate their findings with Fak loss are specific to instances where EGFR can modulate ERK activation, providing strong evidence for their model. Finally, they also correlate their findings in the murine model with patient samples and with trends in the TCGA database. Collectively, these create a solid and convincing basis for their proposed model.

Weaknesses:

(1) The murine data is largely confined to the cecum. While the analysis of the cecum is appropriate based on the cecum specificity of their phenotype, they often use these findings to make broader generalizations about the nature of tumorigenesis in the intestinal epithelia and in CRC more generally. In my opinion, there was insufficient evidence presented supporting the extension of the proposed model beyond the cecum. While this is a weakness, it could be part of a growing effort to characterize left and right-sided malignancies as related but separate disease processes.

(2) The authors generally do a good job of focusing their analysis on the cecum and supporting their model. For example, Figure 5A examines different colon compartments, including the cecum. However, the authors fail to demonstrate that Fak loss only promotes Lrg4 upregulation in the cecum, where they observe an increase in BRAF V600E dysplasia and carcinoma. This is again seen in Figure 6A, where they only characterize Nedd4 expression in the cecum and not other compartments of the colon.

(3) The authors evaluate a broad range of tissues, including normal colonic mucosa, polyps, pre-cancerous dysplastic lesions, adenocarcinomas, and adenocarcinoma cell lines. While this breadth is a strength of the paper, the authors, at times, equate experimental observations in each of these conditions, despite the difference in the biology of these tissues/cells. For example, in their mouse model, they equate the development of dysplastic lesions and carcinoma lesions. This makes it difficult to accurately interpret their data and conclusions.

(4) In Figure 5i, this experiment was only completed in one cell line (HT29), despite the conclusion that Lrg4 expression is increased by decreased ERK phosphorylation due to protein stabilization. HT29 cells are a transformed human CRC cell line, quite different than a pre-malignant cecum intestinal epithelial cell. While convincing, the authors could have performed this key experiment in non-transformed murine cecal organoids (as they did for other experiments in Figure 5E), which would better recapitulate the mouse and pre-malignant setting to explain their mouse phenotype.

(5) While a large portion of the discussion focusses on the therapeutic implications of these findings, the authors only really investigate tumorigenesis. They likely have additional investigations planned for future manuscripts.

---

## [Referee Report · Reviewer #2 (Public Review)]

Summary:

The manuscript by Gao et al. described a study identifying the role of FAK in fine-tuning the activation levels of ERK signaling in BRAF-V600E-driven colorectal cancer. The authors generated new mouse models combining Vill-Cre mediated BRAF-V600E expression with FAK deletion. Analyses of intestinal tumor phenotypes revealed that FAK-loss promotes BRAF-V600E-induced tumor formation, specifically in the cecum. Interestingly, these tumors closely resemble human sessile serrated adenoma/polyps. Using bioinformatics analysis, the authors found that FAK deletion upregulates the intestinal stem cell and fetal-type transcriptomic signatures compared to mice expressing BRAF-V600E alone. In addition, FAK-loss decreases the phosphorylation of ERK whereas it increases the expression of Lgr4 at both mRNA and protein levels. To mechanistically connect FAK-mediated downregulation of ERK and upregulation of Lgr4 in the context of BRAF-V600E mutation, results from biochemical experiments showed that MEK inhibitor treatment decreases the expression of NEDD4, a previously identified ubiquitin E3 ligase of Lgr4, which coincides with increased Lgr4 protein expression both in cells and in vivo. Moreover, the FAK-dependent modulation of ERK signaling is specific to BRAF-V600E-driven tumorigenesis only as knockout of FAK has no effect in Vill-Cre/KRAS-G12D mice. Collectively, the authors proposed a "just right" model in that a tunable FAK expression controls the optimal level of ERK pathway output needed for BRAF-V600E-induced cecal tumor formation.

Strengths:

This study provides new insights into the mechanisms underlying the serrated pathway-driven tumorigenesis in colorectal cancer. The newly established mouse model with compound mutations of BRAF and FAK offers a useful resource for future studies of the serrated pathway. The conclusions of this paper are mostly supported by data.

Weaknesses:

However, some aspects of the paper can be strengthened with additional mechanistically focused experiments.

(1) Some of the conclusions of the paper mainly rely on bioinformatic analyses of RNA-seq data. For example, it has been noted in several places in the paper that the knockout of FAK in Vill-Cre/BRAF-V600E mice does not affect the transcriptional outcome downstream of ERK while ERK phosphorylation levels are decreased. This statement is based on the lack of significant difference in the MAPK signature according to GSEA. However, whereas a significant enrichment of certain pathways can be used as support evidence, the lack of enrichment does not necessarily indicate those pathways are not involved. Other experiments are needed to examine the expression of ERK target genes to confirm. Similarly, the upregulation of fetal stem cell signature in FAK knockout mice needs to be verified using other methods besides GSEA.

(2) According to Figure 5i, the half-life of Lgr4 is around 48 hours in HT29 cells. However, it has been reported by at least two other publications cited in this paper (Ref. 44 and 45) that the half-life of Lgr4 is much shorter. This discrepancy is not explained.

(3) The effect of decreased ERK signaling on NEDD4 expression has only been briefly explored in Figure 6. The mechanisms by which FAK-loss and/or inhibition of MEK/ERK activity regulate NEDD4 expression are currently unclear. Moreover, the levels of NEDD4 expression are only analyzed in one mouse per group in Figure 6a. Quantitative analysis of NEDD4 as well as Lgr4 expression in additional numbers of mice will provide more solid support for the inverse correlation between NEDD4 and Lgr4 proteins. Since MEK inhibitor treatment also increases Lgr4 mRNA expression as shown in Figure 5f-g, the relative contribution of this altered mRNA expression vs. NEDD4L-mediated ubiquitination has not been investigated.

(4) It is an interesting finding that knockout FAK has no effect on KRAS-G12D-driven hyperplasia as shown in Figure 7. However, additional studies are needed to further explore the potential mechanisms by which FAK-loss specifically decreases EGFR/ERK signaling in the context of BRAF-V600E mutation.

---

## [Referee Report · Reviewer #3 (Public Review)]

Summary:

Right-sided colorectal Cancer (CRC) is very different from left-sided CRC. Therefore it is important to model this cancer in mice and find new molecular targets. A broad set of data exists on FAK (Focal Adhesion Kinase) being important in colorectal cancer. However, this has focussed on APC mutant CRC which tends to be left-sided. BRAF mutation is common in right-sided CRC (and is rarely mutated with APC). Therefore the authors have tested whether FAK is important in this context. The authors show that FAK deletion surprisingly accelerates BRAF mutant CRC. Tumours arise in the proximal colon (which recapitulates BRAF mutant right-sided CRC). There are low for Lgr5 and high for foetal programmes. Mechanistically they suggest a pathway from FAK to NEDD4 to Lgr4 may underpin this phenotype.

Strengths:

Strong genetic data from FAK revealed that there is an acceleration of tumourigenesis and mice now develop proximal colon tumours and can be viewed as a good model of right-sided CRC.

The expression data between humans and mice is strong.

Weaknesses:

The functional mechanism of how FAK loss promotes tumourigenesis is still quite correlative. An alternative hypothesis is that it drives inflammation in the proximal colon that drives tumourigenesis.

We still did not know the functional role for LGR4 (loss leads to a loss of paneth cells in homeostasis) so I'm not sure you can hypothesise a stem cell role.

---

## [Author Response]

We thank the editor and reviewers for the time invested in our manuscript and their valuable and insightful critiques. However, we believe that the results justified our conclusions in the manuscript well; therefore, we have decided not to revise it.